# Advances on the Study of Diurnal Flower-Opening Times of Rice

**DOI:** 10.3390/ijms241310654

**Published:** 2023-06-26

**Authors:** Mumei Wang, Minghao Chen, Zhen Huang, Hai Zhou, Zhenlan Liu

**Affiliations:** 1State Key Laboratory for Conservation and Utilization of Subtropical Agro-Bioresources, College of Life Sciences, South China Agricultural University, Guangzhou 510642, China; 15011758956@163.com (M.W.); cminghao_cmh@163.com (M.C.); hz549935163@gmail.com (Z.H.); 2Guangdong Laboratory for Lingnan Modern Agriculture, Guangzhou 510642, China

**Keywords:** rice, *indica*–*japonica* hybrid breeding, DFOT, cell wall, temperature

## Abstract

The principal goal of rice (*Oryza sativa* L.) breeding is to increase the yield. In the past, hybrid rice was mainly *indica* intra-subspecies hybrids, but its yield has been difficult to improve. The hybridization between the *indica* and *japonica* subspecies has stronger heterosis; the utilization of inter-subspecies heterosis is important for long-term improvement of rice yields. However, the different diurnal flower-opening times (DFOTs) between the *indica* and *japonica* subspecies seriously reduce the efficiency of cross-pollination and yield and increase the cost of *indica*–*japonica* hybrid rice seeds, which has become one of the main constraints for the development of *indica*–*japonica* hybrid rice breeding. The DFOT of plants is adapted to their growing environment and is also closely related to species stability and evolution. Herein, we review the structure and physiological basis of rice flower opening, the factors that affect DFOT, and the progress of cloning and characterization of DFOT genes in rice. We also analyze the problems in the study of DFOT and provide corresponding suggestions.

## 1. Introduction

Hybrid breeding is an effective way to increase rice yield, and an *indica* intra-subspecies hybrid can increase the yield by 20% compared to conventional rice [1], accounting for half of the rice planting area and 60% of the total rice production in China [2]. In the past, hybrid rice mainly utilized hybrid vigour between *indica* rice varieties, but it has reached a bottleneck and further yield improvements are difficult to achieve due to the limitations of the genetic diversity in hybrid parent lines [3]. The *indica* and *japonica* rice varieties are two main subspecies differentiated during the domestication process of Asian cultivated rice. Hybridization between *indica* and *japonica* subspecies has stronger heterosis, and the yields of *indica*–*japonica* hybrids are expected to be 30% higher than *indica* intra-subspecies hybrid rice [1]. However, the utilization of *indica*–*japonica* heterosis is restricted by many factors, such as non-compatibility of the hybrids, different growth periods, super-parental late maturity of the hybrid offspring, and non-overlapping diurnal flower-opening times (DFOTs), which have led to a stagnation of *indica*–*japonica* hybrid breeding [4,5]. With the cloning of multiple *indica*–*japonica* hybrid sterility genes, *indica*–*japonica* hybridization has become possible [3,6]. Generally, *japonica* rice is used as a sterile line, while *indica* rice is used as a restorer line. Usually, the DFOT of *japonica* rice is after 12 noon, while that of *indica* rice is earlier, usually around 10 a.m. [7]. The different DFOTs of parental lines significantly reduce the cross-pollination rate and yield, increasing the cost of hybrid seed production, which has become one of the main limiting factors in the development of *indica*–*japonica* hybrid rice breeding [7,8]. Therefore, developing sterile *japonica* rice varieties with an early DFOT through genetic methods can increase hybrid seed production and reduce seed production costs.

High temperature stress in the reproductive period can affect the activity of pollen and the stigma, pollination, fertilization, and grain filling, leading to reducing yields in cereal crops [9,10,11]. Rice is very sensitive to high temperatures during flower opening [12,13]. With global climate change, high temperatures may become more frequent, and the domestication of plant tolerance is one of the most important tasks for adapting to climate variability. It has been reported that there is a significant relationship between rice DFOT and high-temperature tolerance during the rice flower opening period [14]. The spikelets of rice can be sterile under high-temperature stress above 35 °C for 1 h during the flower opening period and the sensitivity of the stigma to high temperatures decreases within 1 h of the flower opening [15]. On the other hand, low temperatures and dew also affect rice pollination. Therefore, there is a significant differentiation in DFOT between *japonica* rice grown at high latitudes and *indica* rice grown at low latitudes, showing significant environmental adaptability. The early DFOT of *indica* rice is beneficial for improving fertility under high-temperature conditions in low latitudes, while the late diurnal flower opening of *japonica* rice is beneficial for improving fertility under low-temperature conditions in high latitudes [16]. In order to reduce the decline in rice yield caused by high-temperature stress, two main methods have been proposed: one is to cultivate rice varieties that can tolerate high temperatures, and the other is to cultivate varieties that flower earlier in the morning. Studies have shown that earlier flower opening by about 1 h can effectively overcome the decrease in rice fertility caused by high-temperature conditions [12,17,18,19,20], thereby reducing the impact of high-temperature stress on the rice yield.

Currently, there are only a few studies on the molecular mechanisms that control the DFOT of rice. Here, we provide a comprehensive review of the structural and physiological basis of rice flower opening and the factors regulating DFOT, as well as the gene cloning that regulates DFOT and the underlying mechanisms of DFOT with the aim of promoting research in this area of rice breeding. Finally, the problems in the study of DFOT are analyzed, and corresponding suggestions are proposed.

## 2. Advances in Research on Rice DFOT

The DFOT of rice, also known as the floret opening time (FOT), refers to the time of spikelet opening during the day after the heading of rice, which is expressed by the time when the flower opening begins or the peak time of flower opening. The trait of a plant’s DFOT is closely related to genetic factors, as well as its environment, which is the result of the adaptation to its growing environment. A similar DFOT of different plants can promote gene flow to increase heterozygosity, while different DFOTs can ensure reproductive isolation before pollination [21].

### 2.1. The Structural and the Physiological Basis of Rice Flower Opening

Rice flower opening involves a series of structural and physiological changes, which have an impact on the DFOT. Therefore, studying the structural and physiological basis of rice flower opening is crucial for understanding the DFOT of rice.

#### 2.1.1. The Structural Basis of Rice Flower Opening

The normal structure of a rice spikelet consists of a pair of sterile lemmas, lemma, palea, a pair of lodicules, six stamens, and a pistil (Figure 1) [22]. The lemma and palea of rice enclose the inner floral organs through interlocking grooves. The two lodicules are symmetrically distributed at the base of the ovary adjacent to the lemma; they are composed of large thin-walled cells and evenly distributed small vascular bundles [4].

The lodicule is the most critical organ that controls the opening and closing of the rice spikelet. The opening and closing of the lemma and palea are the processes of the two lodicules absorbing water and swelling or losing water and shrinking (Figure 2). After the two lodicules absorb water and expand, they push the lemma outward and squeeze the palea inward, causing the interlocking groove of the lemma and palea to loosen, resulting in the opening of the spikelet. When the spikelet is fully opened, the vacuolar membrane of the lodicule cells ruptures, then the cells self-digests its intracellular protein components and organelles. The water and disintegrated substances in the lodicules are transported to the spikelet axis through the vascular bundles, causing the lodicules to lose water. The elasticity of the lemma and palea causes them to close again due to the force of the spikelet axis [4,8,23]. The water absorption and swelling of the lodicules during this process are affected by the degree of relaxation of the lodicule cell wall and the osmotic potential of the cell. Only when the osmotic potential is low and the cell wall is relaxed does the cell have the potential to absorb water and expand. The cell wall plays an important role in supporting and defining the shape of the plant cells and maintaining the cell’s turgor pressure [24,25]. In order to achieve the desired size, the plant cells can increase their osmotic pressure and reduce the expansion pressure of the cell wall [25,26]. The lodicule is mainly composed of thin-walled cells, which only contain primary cell walls. The primary cell wall is a dynamic structure composed of cellulose, hemicellulose, and pectin, which can be adjusted according to the needs of cell growth and development [25,27,28,29,30]. Studies have shown that the degree of methyl-esterification of pectin in the cell wall is positively correlated with the DFOT. Higher pectin methylesterase (PME) activity reduces the degree of methyl-esterification of pectin, increases the degree of calcium binding, makes the cell wall harder, and delays the DFOT. Conversely, lower PME activity increases the degree of methyl-esterification of pectin, reduces the degree of calcium binding, makes the cell wall softer, and advances the DFOT [31]. Pectin can change the extensibility of the cell wall and may also play a critical role in cell wall remodeling [30,32,33].

#### 2.1.2. The Physiological Basis of Flower Opening in Rice

Rice flower opening is the result of a balance between the osmotic pressure and cell wall expansion pressure of the lodicule cells; changes in the metabolites in these cells are crucial for maintaining osmotic pressure. During rice flower opening, the dry weight and soluble sugar content of the lodicule cells increased, resulting in an increase in the osmotic pressure and cell swelling [4,34]. Before flower opening, there is an increase in glycolysis activity within the lodicule cells, leading to the appearance of vacuolar invertase, which can hydrolyze sucrose into glucose and fructose, causing an increase in the osmotic pressure within the cells. In addition, during rice flower opening, the expression of the starch hydrolase genes increases, leading to a significant degradation of starch granules in the interlocking groove and lodicules [35]. This suggests that the increase of soluble sugar content in the lodicules is closely related to glycolysis, vacuolar invertase, and starch hydrolase. The levels of K^+^, Na^+^, Ca^2+^, and Mg^2+^ in the lodicules does not change significantly before or during flower opening, and only decreases after the lodicules began to wither. Moreover, the levels of inorganic ions in the lodicules are much lower than those of soluble sugars, suggesting that inorganic ions play a minor role in regulating the osmotic pressure [4]. However, another study found that the expansion and withering of the lodicules are related to the spatial and temporal dynamics of Ca^2+^ in the lodicules [36]. The day before flower opening, calcium is mainly located in the cell walls of the lodicule epidermal cells. Four hours before flower opening, the calcium disappears from the cell walls and relocates to the cytoplasm of the lodicule epidermal cells. During flower opening, large amounts of calcium are present in the vacuoles of the lodicule epidermal cells. One hour after flower opening, the calcium content decreases. Six hours after flower opening, calcium accumulates in large quantities in the cytoplasm, vacuole membrane, and cell walls of the lodicule epidermal cells again [36] (Figure 3). This indicates that inorganic salt ions may primarily function as signaling molecules in the regulation of flower opening.

Based on the findings above, we summarized some possible genes affecting the DFOT of rice, including sugar metabolism and cell wall synthesis and modification-related genes (Figure 4). These genes affect the DFOT mainly by changing the expansion pressure or osmotic pressure of the lodicule cells to regulate their size, thereby affecting rice flower opening.

### 2.2. The Factors Affecting the Regulation of DFOT

The DFOT of rice is mainly regulated by genetic background, plant growth regulators, and environmental factors (Table 1) [7]; therefore, its regulating mechanism is complex. 

#### 2.2.1. The Influence of Genetic Factors on DFOT

Different rice varieties may have different DFOTs [23]. Generally, the DFOT of the *indica* rice is earlier than that of *japonica* rice, while the DFOT of wild rice is earlier than that of cultivated rice; this is related to their genetic factors. The DFOT of rice is a stable genetic trait, and the genetic factors that cause differences in DFOT mainly include floral organ characteristics, such as the lodicule structure, grain length and width, number of leaf hairs, and length and softness of glume hairs. Compared with male-fertile rice, male-sterile rice lines usually have a dispersed DFOT, and the peak flower opening period is concentrated in the afternoon. Most flower organs of male-sterile rice lines are deformed, and can include abnormal stamens, degraded male organs, and fewer vascular bundles and conduits in the lodicules. This slows water absorption and reduces the elasticity of the lodicules, leading to a delayed DFOT. As a result, hybrid seed production rates are low due to the parents’ non-overlapping DFOTs during seed production [37]. There are also differences between the DFOT of *indica* and *japonica* rice. Among the different grain types in the *indica* rice subspecies, the rounder the grain is, the later its DFOT; conversely, the longer and thinner the grain is, the earlier its DFOT [38]. Long-grain or medium-grain rice varieties, as well as some late *japonica* round-grain rice lines show early DFOT characteristics due to their genetic background of *indica* rice. Therefore, grain length and width have become one of the reference indicators for selecting male-sterile lines with early DFOT traits [39]. It is worth noting that there is a certain correlation between rice DFOT and the number of leaf hairs, glume hair characteristics, and seed staining degree with phenol [40]. The rice varieties with fewer leaf hairs or seeds that are easily stained with phenol will reach the peak flower opening time later, and the varieties with longer, more disordered, and softer glume hairs generally show earlier DFOTs and have a longer diurnal flower opening duration. It is still not clear whether there is a relationship between the DFOT and grain length and leaf hair number. In general, *indica* rice has longer grains and more leaf hairs, while *japonica* rice has shorter grains and fewer leaf hairs. Generally, the DFOT of *indica* rice is much earlier than *japonica* rice; therefore, the correlation between the DFOT and the grain length or leaf hair numbers might be due to the differences in the genetic background of *indica* and *japonica* rice varieties.

#### 2.2.2. The Influence of Plant Hormones and Growth Regulators on DFOT

Plant hormones play an important regulatory role in the opening and closing of rice spikelets. With the development of biotechnology and the increasing demand for hybrid rice seed production, a growing number of plant growth regulators that have been synthesized artificially or are extracted from microorganisms that have similar physiological and biological effects to endogenous plant hormones are being used to solve the problem of non-overlapping DFOTs between the parental lines. Currently, the reported plant growth regulators used to regulate flower opening mainly include methyl jasmonate (MeJA), auxins, “920”, triacontanol, Huaxinling, and others.

Jasmonates (JAs) are a type of fatty acid derivative containing a cyclopentanone basic structure, including jasmonic acid and its various derivatives, such as jasmonoyl-isoleucine (JA-Ile), MeJA, 12-oxo-phytodienoic acid (OPDA), and so on. JAs play an important role in plant growth and development [41,42,43]. The distribution of endogenous JAs in spikelet organs is tissue-specific and development stage-specific. The JA content in rice florets remains stable before flower opening, but sharply increases to a peak during flower opening before declining after flower opening [44]. Before flower opening, the JA level in the pedicel of the spikelet is the highest. During flower opening, the JA levels in the stamens and lodicules are significantly higher than in other floral organs, while the JA content in the pedicel is the lowest. Consistent with the changes in JA levels, the expression of JA biosynthesis-related genes *OsDAD1*, *OsAOS1*, *OsAOC*, *OsJAR1*, and *OsOPR7* in the stamens and lodicules increases significantly, while their expression in the pedicel is significantly down-regulated. The *OsOPR7* gene has been reported to affect carbohydrate transport in rice lodicules during flower opening [45,46]. The expression of JA signal transduction pathway-related genes *OsCOI1b* and *OsJAZs* also increases significantly in the lodicules [45]. The reduction in JA content in the lodicules of the cytoplasmic male sterile (CMS) line Zhenshan 97A delayed and dispersed the DFOT of spikelets, indicating that endogenous JA regulates the opening and closing process of rice spikelets [8]. MeJA is an important DFOT regulator in hybrid rice seed production. Spraying MeJA can induce spikelet opening in *japonica* CMS lines. The effects of MeJA on inducing flower opening in CMS lines are more sensitive than in fertile lines [47,48]. MeJA can also induce flower opening in *indica* CMS lines with a significant flower opening peak, showing that MeJA has universal applicability in regulating DFOT in hybrid rice production [49]. Exogenous MeJA has also been reported to promote the DFOT of sorghum, and the promotion effect increases with the increasing concentration of exogenous MeJA [50]. Therefore, JAs play an important role in crop breeding.

Auxin has been reported to be associated with plant flower opening [51,52]. Exogenous auxins, indole-3-acetic acid (IAA), and naphthaleneacetic acid (NAA) can delay the DFOT of rice. The content of IAA in rice spikelets rapidly decreased within 2 h before natural opening, and the expression of IAA biosynthetic genes (*OsTAR2*, *OsYUCCA3*/*4*/*8*) in the spikelets decreased correspondingly when the spikelets opened, while the genes encoding enzymes that catalyze IAA conjugation (*OsGH3.2*, *OsGH3.8*), IAA efflux carrier genes (*OsPIN1*, *OsPIN1a*), and the gene encoding the positive regulator of the IAA polar transport factor BG1 were all significantly up-regulated [53,54]. This indicates that the DFOT of rice is negatively regulated by auxins.

Gibberellins are widely distributed plant hormones that have important effects on plant growth, development, flower opening, and fruiting [7]. The “920” additive is a mixture of various homologs of gibberellins, and exogenous application of “920” can promote the flower opening of the female parent, increase the chance of overlapping DFOTs with the male parent and the hybrid seed production rate in rice [55]. Spraying “920” before flower opening of rice sterile lines can induce an earlier DFOT by 0.5~1 h, increasing the rate of overlapping DFOTs between parents to 15–20% and improving seed production rates [56]. Currently, “920” is widely used in rice plantation to increase the chance of overlapping DFOTs between parents and the pollination rates, and then improve seed production yields.

Triacontanol, also known as melissyl alcohol, is a natural long-chain aliphatic alcohol. As a plant growth regulator with no toxic effects, triacontanol can promote the growth and development of plant roots, stems, leaves, shoots, and flowers, and enhances various physiological functions, such as increasing the chlorophyll content and enhancing photosynthesis [7]. Triacontanol can promote peak flower opening of sterile lines, but has almost no effect on maintainer lines in rice. Spraying the sterile line and the maintainer line at the same time with triacontanol can bring the peak flower opening times of the two lines closer and increases the rate of overlapping DFOTs [57]. A physiological analysis shows that triacontanol can increase the activity of photosynthetic phosphorylation and promote the accumulation of adenosine triphosphate (ATP), providing energy for the flower opening of sterile lines [58].

Huaxinling is a type of plant growth nutrient that can promote the early flower opening of parent plants and can also promote the vegetative growth and reproductive development of rice. It is a newly refined and highly effective DFOT regulator that has the effect of promoting early flower opening of the female parent and increasing the outcrossing rate [59]. Huaxinling can increase the rate of female parent flower opening before noon by about 36.25%; it can also prolong the duration of peak flower opening of the parents, thereby partially solving the problem of non-overlapping DFOT between the parent lines [60].

In summary, a certain concentration of MeJA, “920”, tricontanol, Huaxinling, and other plant growth regulators can promote the flower opening of the male sterile lines and cross-pollination rate, increasing the yield of hybrid rice seed production. However, exogenous plant growth regulators have certain limitations in terms of concentration and application time, and the growth status of plants in different environmental conditions is also different. Improper utilization of growth regulators may not achieve the expected results and may also increase the cost of hybrid seed production. Therefore, it is critical to promote early flower opening in rice through genetic means in hybrid seed production.

#### 2.2.3. The Influence of Environmental Factors on DFOT

Plants are sessile organisms and cannot escape extreme environments. They can only adapt to constantly changing conditions by adjusting themselves. There is significant differentiation between the flower opening habits of *indica* and *japonica* rice. These differences are related to their growth environment. Most of the *japonica* rice varieties grow in high latitude areas, where the morning temperature is relatively low, resulting in later DFOTs. On the other hand, *indica* rice mainly grows in low latitude areas where the temperature is relatively high; therefore, it needs to flower earlier to avoid high temperatures. In fact, the opening of rice spikelets is regulated by various external factors, including temperature, light, CO_2_ concentration, humidity, etc. Currently, temperature is considered to be the most important factor affecting the DFOT, followed by light and CO_2_ [61].

Spikelets of rice are very sensitive to external temperatures. For the same rice variety, the DFOT can be different on two days with different temperatures. Flower opening occurs more frequently and reaches its peak quickly under high temperatures, while it is delayed or does not occur under low temperatures [62]. The average daily temperature affects the DFOT of rice. High temperatures facilitate early flower opening, but excessively high or low temperatures affect the overlapping of the DFOTs of the parents [40]. A high temperature during the flower opening period mainly changes the opening dynamics and physiological characteristics of the rice spikelets, reduces the anther dehiscence rate and pollen vitality, and lowers the seed setting rate, resulting in reduced rice production [16,63]. Therefore, raising the temperature appropriately in the morning can promote flower opening of rice, while reducing the temperature can delay it [15]. However, there is no research reported on the regulatory mechanism of rice DFOT in response to temperature changes, and the genetic basis of the different temperature response characteristics of *indica* and *japonica* rice is still unclear. The length of daylight and light intensity also affect the DFOT of plants [64,65]. Studies have found that rice spikelets open at specific times of the day under natural conditions, while the DFOT of rice spikelets is disrupted and becomes scattered, even lasting for a whole day under artificially continuous light conditions [66]. Additionally, studies have shown that increased solar radiation before flower opening can advance the DFOT [15,61].

CO_2_ can effectively induce water absorption and swelling of lodicules and promotes early opening of rice spikelets. Treatment with CO_2_ gas with a concentration greater than 5% or a CO_2_ aqueous solution with a pH less than 5.6 for about 20 s can significantly promote spikelet opening. However, increasing the treatment time or CO_2_ concentration cannot further increase the number of opening spikelets. Treating spikelets with 0.5% NaN_3_ or 0.5% KCN, which are respiratory inhibitors, inhibits cell respiration and reduces CO_2_ concentration in spikelets, subsequently inhibiting spikelet opening [4]. Some studies suggest that the increase in CO_2_ concentration enhances respiration and causes an increase in spikelet temperature, which promotes spikelet opening [67]. Further studies have found that organic acids, such as formic acid, acetic acid, and propionic acid with a volume fraction of about 1%, can also significantly promote rice flower opening. It is speculated that lower cell pH promotes cell wall relaxation, thus inducing flower opening [4,68].

The influence of humidity on rice flower opening is still unclear. Some studies suggest that humidity has no effect on rice flower opening, while others show that appropriate humidity is needed for flower opening. It has also been demonstrated that high humidity is more conducive than high temperature for male sterile lines to flower early and enter the flower opening peak, while delaying flower opening of the restoring line parent [15,69,70]. However, there is no consensus on the effect of humidity on flower opening.

### 2.3. Cloning and Molecular Mechanism Studies of DFOT Genes

Generally, the DFOT trait is considered to be a quantitative trait that is controlled by a quantitative trait locus (QTL). In order to clone the genes affecting DFOT and explore the genetic mechanism of DFOT, genetic populations were constructed by using different rice varieties with different DFOTs. Some QTLs controlling the DFOT of rice have been mapped, and one DFOT gene has been cloned and characterized (Figure 5 and Table 2).

The study by Wang et al. suggests that the early DFOT of *indica* rice BC-801 is controlled by one dominant gene through genetic analysis of *indica* and *japonica* rice [76]. Chen et al. found that DFOT may be controlled by multiple genes, with incomplete dominance in F_1_ and super-parental effects in F_2_. The genes controlling DFOT are not linked to the genetic factors related to the differentiation of *indica* and *japonica* subspecies [77]. Bai [71] mapped three QTL loci controlling rice DFOT on chromosomes 5 and 10 using the recombinant inbred line population constructed by hybridizing Chuanxiang 29B with Lemont and found a contribution rate as high as 73.72% on chromosome 5. Thanh et al. compared the flower opening characteristics of Asian wild rice (*O. rufipogon* W630, early flower opening) and *japonica* rice (*O. sativa japonica* cv. Nipponbare, late flower opening), and mapped three QTL loci controlling DFOT on chromosomes 4, 5, and 10 by conducting QTL analysis using the BC_2_F_8_ backcross population [72]. Ma et al. mapped six QTL loci on chromosomes 1, 2, 7, 8, 10, and 12 using the hybrid-derived recombinant inbred line population developed by a cross between the *indica* rice variety "Qiuguang" and the *japonica* rice variety "Qishanzhan" [40]. Wan et al. mapped four QTL loci on chromosomes 1, 10, and 12 using the F_2_ population generated by hybridization between the early DFOT *indica* rice variety WAB368-B-2-H2-HB and the late DFOT *indica* rice variety Liuqianxin [73]. Phuong et al. mapped one QTL locus between SSR markers RM13 and RM574 on chromosome 5 and another QTL locus between RM44 and RM223 on chromosome 8 using a backcross population between W630 and *indica* rice IR36. The homozygote lines of these identified loci have the potential to promote earlier flower opening [74]. Hirabayashi et al. identified a stable QTL locus, *qEMF3*, on chromosome 3, which advances the DFOT by 1.5–2.0 h in both the temperate rice variety Nanjing 11 of Japan and the tropical rice variety IR64 of the Philippines [17]. This locus may alleviate the heat stress-induced spikelet sterility under high temperature conditions. Additionally, some genes regulating rice DFOT have also been reported. We identified a highly expressed gene *DFOT1* (*Diurnal Flower Opening Time 1*) using the transcriptome data of the Zhonghua 11 (*japonica* cultivar) lodicules collected at different time points during flower opening. Xu *et al*. obtained an early flowering mutant *emf1* (*early-morning flowering1*) by screening the ethyl methanesulfonate mutagenized population of Yixiang 1B with a late DFOT, and they cloned the *EMF1* by map-based cloning and MutMap analysis. *EMF1* and *DFOT1* is the same gene. *DFOT1*/*EMF1* is the first reported gene that directly regulates rice DFOT, and has potential applications in hybrid breeding, providing new ideas for hybrid parent creation and improvement [31,75].

## 3. Discussion and Prospects

### 3.1. Cloning of Early DFOT Genes Promotes the Development of Indica–Japonica Hybrid Breeding

Yuan [78] classified hybrid rice breeding into three stages based on the level of utilization of heterosis: intervarietal, intersubspecific, and intergeneric hybridization. At present, the yield of hybrid rice between the varieties has been difficult to improve. *Indica*–*japonica* hybrid breeding entered the stage as a result. Intersubspecific hybrid rice has the potential to increase the yield by 30% compared to intervarietal hybrid rice [1]. The direct utilization of heterosis between *indica* and *japonica* rice has been achieved by using a widely compatible gene [3,6] and the sterile lines for two-line hybrid rice production [31,79,80,81]. However, the yield of *indica*–*japonica* hybrid combinations in hybrid seed production is generally low. The yield of *indica*–*japonica* hybrid seed production typically ranges from 0.075 to 0.75 t·hm^−2^, whereas the yield of intra-subspecies hybrid seed production within the *indica* variety can reach up to 2.25–3 t·hm^−2^. The reasons include differences in suitable weather conditions for flower opening, heading stages, and DFOTs between the parents, and the different DFOTs of the parents is the main factor [39]. There are differences in the DFOTs of different rice varieties, especially between *indica* and *japonica* rice. Studies have shown that the flower opening peak of *japonica* male sterile lines and *indica* restorer lines differs by about 2.5 h [76], which severely restricts the yield of inter-subspecific hybrid seed production and has become the main limiting factor for the development of *indica*–*japonica* hybrid rice. In addition, high summer temperatures in tropical and subtropical regions as a result of the global climate warming are becoming a major threat to rice production due to the exposure to heat stress during the flower opening period, which may result in spikelet sterility. Therefore, advancing or delaying the DFOT to avoid the hottest time of the day can significantly improve rice’s heat tolerance [14,16].

Because there are many factors that affect DFOTs in rice, including genetic factors, environmental factors, and plant hormones, the mechanism regulating DFOT is complex. At present, there are only two methods that can be used to minimize the difference of the DFOTs between parent lines in hybrid rice production. The first method is by spraying exogenous hormones. Zeng et al. found that MeJA can induce rapid and abundant flower opening in male-sterile lines, leading to early flower opening peak [82]. MeJA also significantly increases the stigma exertion rate of the male sterile lines, thereby improving the hybridization success rate and seed yield. However, exogenous MeJA can also cause seed shedding from the panicle after pollination [83]. Exogenous MeJA not only affects the flower opening of male-sterile lines, but also alleviates the stress caused by high-temperature (≥35 °C). When exposed to high temperatures, male-sterile rice varieties exhibit problems such as non-heading and abnormal fertilization of the female reproductive organs (pistils) [84,85,86,87], leading to a significant reduction in seed yield. Xu et al. found that spraying MeJA on male-sterile lines under high-temperature stress significantly increased the flower opening rate compared to the treatment of spraying water under high-temperature conditions, and the flower opening rate and the stigma exertion rate also greatly increased, thereby significantly improving the seed setting rate and reducing the adverse effects of high-temperature stress on the seed yield [82,88]. Although using MeJA and other similar compounds can significantly increase the yield of hybrid rice in actual production, large-scale use can result in increased production costs. The second method is to clone the DFOT-related genes to carry out genetic improvement, and cultivate rice varieties with earlier DFOT. Zhang et al. [23] found that the *japonica* CMS line Zhe 08A that was backcrossed from Zhe 04A only flowers 20 min earlier than Zhe 04A, but its outcrossing rate reaches as high as 30%, and its seed yield increases from 0.75–1.5 t·hm^−2^ to 1.8 t·hm^−2^ when hybridized with Zhehui F1015. Therefore, developing rice varieties with earlier DFOTs can effectively reduce the production costs of hybrid seeds, promote the development of hybrid breeding of *indica* and *japonica* rice, and effectively ensure food security.

### 3.2. Prospects of the Research of Rice DFOT Genes

In the previous studies, a dozen QTLs controlling the DFOT in rice were mapped by different research groups using forward genetics [17,40,72,73,74]. However, specific genes corresponding to these QTLs have not been cloned yet. Our research group attempted to clone the DFOT genes through forward genetics, but found it very difficult, mainly due to the instability of the DFOT phenotypes in changing environmental conditions. The DFOT is mainly regulated by the balance between the cell wall swelling pressure and cytoplasmic osmotic pressure, which are easily affected by the surrounding environment. These two process-related genes can be investigated through reverse genetics whether or not they are related to DFOT. The genes *DFOT1* and *PMEs* cloned by our group through reverse genetics affect the cell wall structure, which in turn affects the DFOT of rice. In addition, JA has a significant effect on flower opening; therefore, JA pathway-related genes can be further investigated. However, the DFOT genes cloned by reverse genetics may have some negative effects and thus may not be suitable for hybrid rice breeding. For example, knock-out of *DFOT1* can lead to a two-hour advance in the DFOT. However, there are some problems with the knock-out mutants of *DFOT1*, such as non-concentrated DFOT, a DFOT that is too early, and susceptibility to low temperatures in the second half of the year. Therefore, a combination of forward and reverse genetics can be used to clone the DFOT genes. We can identify the genes controlling the DFOT through reverse genetics, and then select superior alleles to verify their inheritance, and finally apply the genes conferring early and concentrated DFOTs to hybrid seed production.

## Figures and Tables

**Figure 1 ijms-24-10654-f001:**
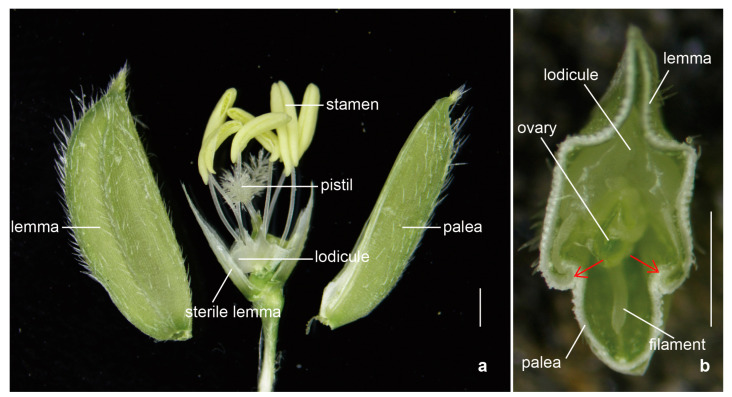
The spikelet structure in rice. The morphological anatomy (**a**) and the cross-section (**b**) of the spikelet in rice. The red arrows point to the interlocking grooves. Bars: 1 mm.

**Figure 2 ijms-24-10654-f002:**
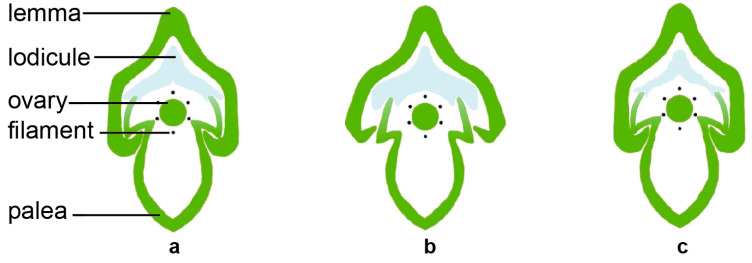
The dynamic process of flower opening and lodicule size in the cross-section of rice spikelet. Unopened flower (**a**), opening flower (**b**), and opened flower (**c**).

**Figure 3 ijms-24-10654-f003:**
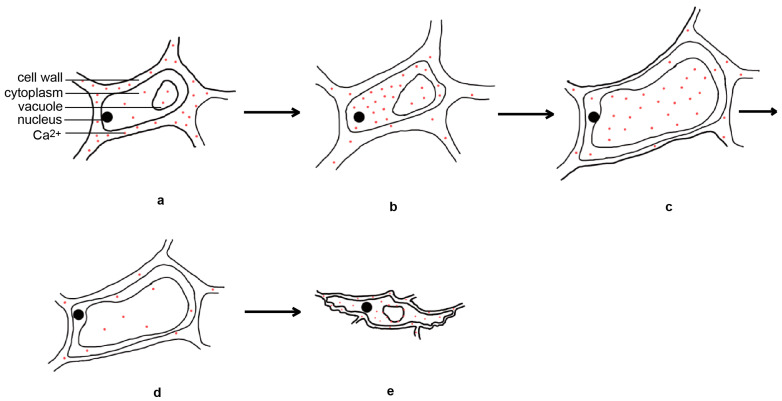
Temporal and spatial dynamical changes of calcium ions in lodicule epidermal cells during rice flower opening and closing processes. The day before flower opening (**a**), four hours before flower opening (**b**), during flower opening (**c**), one hour after flower opening (**d**), and six hours after flower opening (**e**).

**Figure 4 ijms-24-10654-f004:**
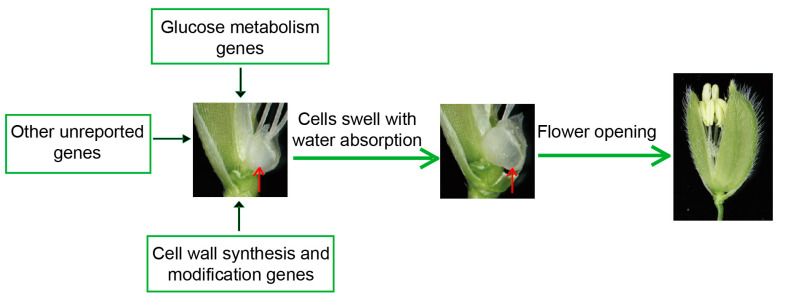
The regulatory mechanism of rice flower opening. The red arrows point to lodicules.

**Figure 5 ijms-24-10654-f005:**
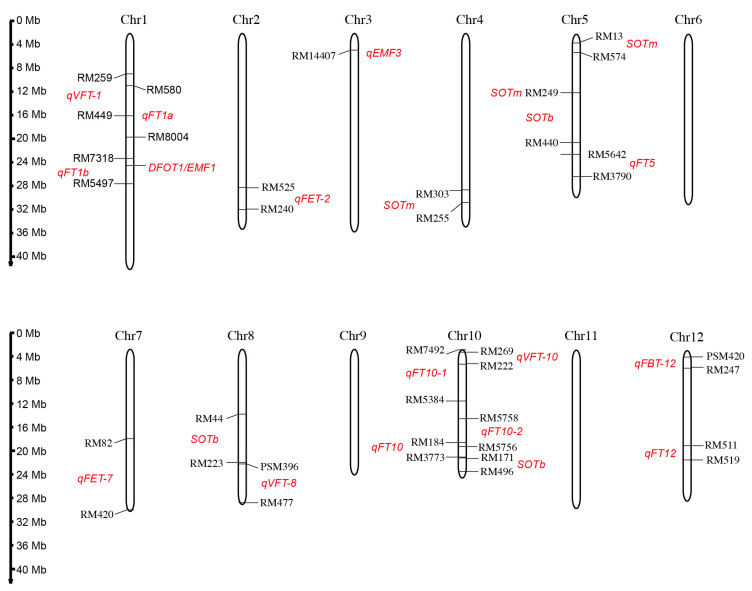
The chromosome location of the identified QTL loci or cloned gene controlling rice DFOT. Red text denotes the putative regions of QTL and the mapped QTL names.

**Table 1 ijms-24-10654-t001:** Main factors affecting the opening of rice flowers.

Main Factors	Classification	Effects
Genetic factors	Varietal differences	Flower opening of wild rice was earlier than that of cultivated rice, and that of *indica* rice was earlier than that of *japonica* rice
	Lodicules’ structure	Sterile line with anomalous lodicule structure has a late DFOT
Plant hormones and growth regulators	Methyl jasmonate (MeJA)	Promote flower opening
	Auxins	Inhibit flower opening
	Gibberellins	Promote flower opening
	Triacontanol	Promote flower opening
	Huaxinling	Promote flower opening
Environmental factors	Temperature	A suitable high temperature promotes flower opening
	Light	Prolonged light exposure disturbs flower opening
	CO_2_	Promote flower opening
	Humidity	Uncertain

**Table 2 ijms-24-10654-t002:** Identification of QTL loci and gene cloning of rice DFOT.

Parents (QTL or Gene Source)	Chromosome	Interval/Gene Locus	Reference
Chuanxiang 29B × Lemont.	5, 10, 10	RM3790-RM5642, RM7492-RM5384, RM5758-RM5756	[71]
W630 × Nipponbare	4, 5, 5,10	RM303-RM255, RM249-RM440, RM249, RM171-RM496	[72]
“Qiuguang” × “Qishanzhan”	1, 2, 7, 8, 10, 12	RM259-RM449, RM240-RM525, RM82-RM420, RM477-PSM396, RM269-RM222, PSM420-RM247	[4,40]
WAB368-B-2-H2-HB × “Liuqianxin”	1, 1, 10, 12	RM580-RM8004, RM7318-RM5497, RM184-RM3773, RM511-RM519	[73]
W630 × IR36	5, 8	RM13-RM574, RM44-RM223	[74]
EMF20 × Nanjing 11 (NJ)	3	RM14407	[17]
EMF1/DFOT1 was cloned from Yixiang 1B and ZH11	1	Os01g0611000	[31,75]

## Data Availability

Not applicable.

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
