# Peer review of "Advances on the Study of Diurnal Flower-Opening Times of Rice"

_ijms, 2023, doi:10.3390/ijms241310654_

Round 1

Reviewer 1 Report

Overall, this manuscript cannot be considered as a good reference for the people who are interested in hybrid rice production or DOFT due to its brief nature and less informative contents. writing style is not methodical because information is not given in a structural manner. There were some English style issues and I hope the authors will run an English style test. I recommend a deeper analysis and examination of contents and expand the discussion section for more valuable conclusions.

To be accepted to IJMS, consideration of several major comments below is required.

1. Introduction

This section loosely explains the background of hybrid rice production methods and materials. However, more information such as currently cultivated rice hybrids and their parents along with time of flower opening can be helpful to a reader who is interest in DOFT. This section needs more upgrades. perhaps using a map can be more fancy.

hybrid rice production related statistics can be represented by a graph. Please consider.

line 125: Flowering or flower opening? it’s confusing with heading and anthesis.

2. Advances in Research on Rice DFOT

This section is relatively well written, however can be improved with deeper molecular mechanism and figures.

2.1.2. The physiological basis of flowering in rice

The authors have provided complex explanation for the physiology of flower opening. However, a figure explaining the process can be less complicated. Figures may include the physiological processes and corresponding genes. calcium levels can also be depicted by a graph in a time series figure.

2.2. The factors affecting the regulation of DFOT

Authors has done a good survey but then again, explained in a complicated way. Its recommendable to summarize the factors, variations and effect under a single table for simplification.

line 170:

Grain and leaf characters has a correlation with DOFT. Please elaborate the causation. without proper causation, this statement can be unusable. please discuss the causations, possible solutions in discussion section.

line 213:

Are you suggesting JA is important for producing efficient hybrids by facilitating crosspollination or JA related alleles are necessary to breed compatible DOFT rice varieties? Please explain.

line 296:

There is a contradiction between line 270. Please elaborate how increasing temperature can open spikelet without damaging the spikelet.

Table 1:

This table must be separated further. i.e. QTL/gene name, chromosome, function and reference. method and details are not necessary. Please try to show a figure with chromosomes and locations.

Figure 4. must have more information other than the text already explained the mechanism. Please remove because its redundant.

3. Discussion and Prospects

Discussion segment must not repeat the contents from above sections, it’s should discuss the ongoing and possible studies which can help to understand the problem in hand.

Most contents are repeated from introduction or must be moved to introduction.

Line 362-376: these contents were already stated in the introduction. Please remove. 

There were some English style issues and I hope the authors will run an English style test.

Author Response

First of all, thank you very much for your comments. We have been modified the manuscript according to your suggestions. Please see the attachment.

Reviewer 2 Report

The Authors present an interesting Review concerning the breeding of rice (Oryza sativa) in order to increase the yield and they underline the strategy of modifying/regulating the diurnal flower opening time of the subspecies to favor cross-pollination. The topic is very interesting considering the economic repercussions and the need to select hybrids that can adapt to climate change. The Authors have considerable experience in this research sector as evidenced by their publications included in the bibliography and what they report in the paper. The review of the scientific literature concerning this sector is in-depth and updated and the Authors adequately examine all the various factors that influence the DFOT and the methods to modify it, highlighting benefits and defects/contraindications. It should be appreciated that a review talks about future prospects. In my opinion the paper is suitable for the publication.

Only some notes:

Please check that all the abbreviations are explained at the first citation

   Please check the space at lines 30-31, maybe a system error during the building of the pdf

   Please attention to pelea at the beginning of line 99
     Line 188 “acontanol, and Huaxinling etc.” is acontanol and Huaxinling? Or acontanol, Huaxinling and etc
-    ?Maybeyou can change in "effect"s at this phrase at line 208? “the effect of MeJA on inducing spikelet opening in CMS lines are more sensitive than in fertile lines”. The verb is plural.

The English is good, only a minor control

Author Response

First of all, thank you very much for your comments. We have been modified the manuscript according to your suggestions.

Point 1: Please check that all the abbreviations are explained at the first citation

Please check the space at lines 30-31, maybe a system error during the building of the pdf.

Response 1: Thanks for the reviewer’s reminder. We have made changes in the revised manuscript.

Point 2: Please attention to pelea at the beginning of line 99.

Response 2: Thanks for the reviewer’s reminder. We have made changes in the revised manuscript.

Point 3: Line 188 “acontanol, and Huaxinling etc.” is acontanol and Huaxinling? Or acontanol, Huaxinling and etc.

Response 3: Thanks for the reviewer's reminder. We have changed it to "acontanol, Huaxinling and etc" in the revised draft.

Point 4: Maybe you can change in "effect"s at this phrase at line 208? “the effect of MeJA on inducing spikelet opening in CMS lines are more sensitive than in fertile lines”. The verb is plural.

Response 4: Thanks for the reviewer's reminder. We have made changes in the revised manuscript.

Point 5: Comments on the Quality of English Language

The English is good, only a minor control.

Response 5: Thanks for the referee’s comments. We have made changes in the revised manuscript.

Please see the attachment for the revised manuscript.

Thank you again for your comments. I wish you a happy life!

Round 2

Reviewer 1 Report

The revised version has been updated to mostly reflect the reviewer's questions.

The revised version has been updated to mostly reflect the reviewer's questions.